# Chitosan–Polyvinyl Alcohol Nanocomposites for Regenerative Therapy

**DOI:** 10.3390/polym15234595

**Published:** 2023-12-01

**Authors:** Carlos David Grande-Tovar, Jorge Ivan Castro, Diego López Tenorio, Paula A. Zapata, Edwin Florez-López, Carlos Humberto Valencia-Llano

**Affiliations:** 1Grupo de Investigación de Fotoquímica y Fotobiología, Universidad del Atlántico, Carrera 30 Número 8-49, Puerto Colombia 081008, Colombia; 2Tribology, Polymers, Powder Metallurgy and Solid Waste Transformations Research Group, Universidad del Valle, Calle 13 No. 100-00, Cali 76001, Colombia; jorge.castro@correounivalle.edu.co; 3Grupo Biomateriales Dentales, Escuela de Odontología, Universidad del Valle, Calle 4B # 36-00, Cali 76001, Colombia; diego.lopez.tenorio@correounivalle.edu.co (D.L.T.); carlos.humberto.valencia@correounivalle.edu.co (C.H.V.-L.); 4Grupo de Polímeros, Facultad de Química y Biología, Universidad de Santiago de Chile, Santiago 9170020, Chile; paula.zapata@usach.cl; 5Grupo de Investigación en Química y Biotecnología QUIBIO, Universidad Santiago de Cali, Calle 5 No. 62-00, Cali 760035, Colombia; edwin.florez00@usc.edu.co

**Keywords:** clove essential oil, chitosan composites, nanobioglass, subdermal tissue regeneration, scaffolds

## Abstract

Tissue accidents provide numerous pathways for pathogens to invade and flourish, causing additional harm to the host tissue while impeding its natural healing and regeneration. Essential oils (EOs) exhibit rapid and effective antimicrobial properties without promoting bacterial resistance. Clove oils (CEO) demonstrate robust antimicrobial activity against different pathogens. Chitosan (CS) is a natural, partially deacetylated polyamine widely recognized for its vast antimicrobial capacity. In this study, we present the synthesis of four membrane formulations utilizing CS, polyvinyl alcohol (PVA), and glycerol (Gly) incorporated with CEO and nanobioglass (n-BGs) for applications in subdermal tissue regeneration. Our analysis of the membranes’ thermal stability and chemical composition provided strong evidence for successfully blending polymers with the entrapment of the essential oil. The incorporation of the CEO in the composite was evidenced by the increase in the intensity of the band of C-O-C in the FTIR; furthermore, the increase in diffraction peaks, as well as the broadening, provide evidence that the introduction of CEO perturbed the crystal structure. The morphological examination conducted using scanning electron microscopy (SEM) revealed that the incorporation of CEO resulted in smooth surfaces, in contrast to the porous morphologies observed with the n-BGs. A histological examination of the implanted membranes demonstrated their biocompatibility and biodegradability, particularly after a 60-day implantation period. The degradation process of more extensive membranes involved connective tissue composed of type III collagen fibers, blood vessels, and inflammatory cells, which supported the reabsorption of the composite membranes, evidencing the material’s biocompatibility.

## 1. Introduction

Regenerative medicine seeks to restore tissues by incorporating cells or growth factors that facilitate recovery [1]. It is an efficient technology that can be improved with the development of more advanced technologies that allow scaffolds to be appropriately designed so that they integrate correctly with the cellular environment and can incorporate cellular material to improve their interactions with the biological environment, including the formation of complex vascularization [2,3].

To repair and replace injured tissues, tissue transplants from the same patient (autografts) or other individuals (transplant or allograft) are used. However, these treatments have problems associated with their application [4]. In the case of autografts, there are severe anatomical limitations at the harvesting site; in addition, they can be painful and expensive and generate hematomas and infection. On the other hand, allografts present limitations regarding sufficient availability of tissue and the risk of diseases that cause rejection or even death of the patient. Therefore, tissue engineering seeks to regenerate damaged tissues instead of replacing them with structures that can help restore, maintain, or improve tissue function.

Typically, scaffolds are designed from synthetic and natural polymers, which offer various advantages and disadvantages during use. Biopolymers have the advantage of being biocompatible and biodegradable, as well as stimulating interaction with the cellular biological environment without generating aggressive responses. However, they have poor mechanical and barrier properties [5]. On the other hand, synthetic polymers have better mechanical performance and economic production, although they are not biodegradable, nor do they stimulate cell adhesion or proliferation in many cases. Therefore, a beneficial strategy to mitigate the deficiencies of both polymers is to create physical mixtures of them [6]. Among the synthetic polymers most often used for the design of scaffolds is polyvinyl alcohol (PVA), thanks to its thermal and mechanical stability, biocompatibility, flexibility, and low cost [7]. On the other hand, among the most used biopolymers for biomedical applications is chitosan, a heteropolysaccharide composed of glucosamine and β-1,4-N-acetylglucosamine units, with excellent biocompatibility and stimulating capacity for the development of new tissues [8].

On the other hand, another strategy used to improve the bioactivity of scaffolds is to generate porous three-dimensional nanocomposites by incorporating nanomaterials in the polymer mixtures [9], which favors their integration into the body tissues [10]. Bioactive materials are sometimes included to mimic the host tissue’s biocompatibility, porosity, and architecture [11].

Among the most interesting bioactive materials are nanobioglasses (n-BG), which stimulate a calcium and phosphate layer and advance angiogenesis on the surface of the scaffolds without producing an adverse immune reaction [12,13,14,15]. Among the n-BGs that can be used for biomedical applications are 1393BG and 45S5, the compositions of which bind different minerals such as SiO_2_, Na_2_O, CaO, P_2_O_5_, and others. However, the most used is 45S5, because its composition does not allow for the production of necrotic tissue or pus [16,17]. The main disadvantage is the adequate control of porosity, which requires specific conditions; therefore, these conditions can affect the nucleation phenomena that are ideal for biomedical procedures [18,19,20]. Our group previously published a work demonstrating the advantage of incorporating n-BG type 45S5 in polylactic acid (PLA) scaffolds to improve their biocompatibility and reinforce their thermal and mechanical resistance [21]. 

On the other hand, significant benefits have also been reported in the design of nanocomposite scaffolds incorporating essential oils, such as a higher rate of degradation, greater porosity and biocompatibility, and lower inflammatory response [22,23]. Essential oils are complex, odorous liquid mixtures of highly volatile compounds obtained from the secondary metabolism of plants by solvent extraction or hydrodistillation, with excellent anti-inflammatory, antimicrobial, antiviral, and therapeutic properties [24].

*Syzygium aromaticum* L. (clove, *Myrtaceae* family, CEO) is widespread in Madagascar, Sri Lanka, Indonesia, and China. It is a plant that produces an essential oil with antioxidant, insecticidal, antibacterial, and antifungal properties, and is classified as safe for consumption (GRAS) by the Food and Drug Administration (FDA), facilitating its application in the cosmetic, food, and perfumery fields [25]. According to previous reports, the oil comprises at least 50% eugenol and 10–40% eugenol acetate, β-caryophyllene, and α-humulene [26]. Among the fascinating biological properties that have been observed of clove essential oil (CEO) for biomedical applications are the anti-inflammatory activity [27], analgesic [28], anesthetic [29], anticancer [30,31,32], and other biological activities [25] that promote tissue regeneration. Pd nanoparticles have been made together with the CEO for evaluation as cytotoxic agents against cervical cancer; it was demonstrated that this material is biocompatible, as it induced a concentration-dependent inhibition against HeLA cells that could have occurred due to the pharmacologically active compounds retained on the surface of the PdNPs of the clove bud extract [33].

Despite all the advantages of CS-PVA composites that have previously been reported, including CS/PVA/TTEO tea tree essential oil, which accelerates and stimulates subdermal tissue regeneration after 90 days of implantation without necrotic tissue [34], the application of scaffolds based on CS/PVA/CEO/GLY/n-BG composites has not been reported. The incorporation of n-BG and CEO stimulates rapid tissue regeneration and reduces tissue inflammation. It avoids microbial contamination, making it an excellent candidate for soft and hard tissue regeneration. Therefore, in this work, we set out to prepare scaffolds based on CS/PVA/CEO/GLY/n-BG for the renewal of subdermal tissue of biomodels (5-month-old male Wistar rats), seeking a faster regeneration of tissue in the absence of fibrous capsules that is applicable in tissue engineering and therapeutic medicine.

## 2. Materials and Methods

### 2.1. Materials

All the reagents used in this study were obtained from Sigma, Aldrich. The chitosan (CS) used had a low molecular weight (1 × 10^6^ Da), a deacetylation degree of >80%, and a viscosity of 20–300 cP (in 1% acetic acid), according to the provider, while the polyvinyl alcohol showed 90% hydrolysis and contained 95,000 g/mol.

### 2.2. Synthesis of n-BG

The synthesis and characterization of n-BG with a ternary system (54%-SiO_2_:40%-CaO:6%-P_2_O_5_) have been reported elsewhere [21].

### 2.3. Composition of Clove Essential Oil (CEO)

The clove essential oil (*Syzygium aromaticum* L.) was obtained from Marnys (Madrid, Spain). Essential oil chemical characterization was performed using gas chromatography mass-coupled (GC-MS, Appendix A). For that, an AT 6890 series plus gas chromatograph (Agilent Technologies, Palo Alto, CA, USA) with a mass selective detector (Agilent Technologies, MSD 5975), Column DB-5MS (J & W Scientific, Folsom, CA, USA), 5% -Ph-PDMS was employed. Identification was based on RI (retention indexes) from the Adams (Wiley 138) and NIST05 (Agilent, Santa Clara, CA, USA) databases.

### 2.4. Preparation of the CS/PVA/CEO/GLY/n-BG Composites

Based on our previous synthetic pathway, we prepared the CS/PVA/CEO/GLY/n-BG nanocomposite membranes [22]. All the nanocomposite formulations, with final total solid contents of 4%, were prepared using the drop-casting method. The N-BG was ultrasonicated using an ultrasonic bath (Branson, Madrid, Spain) in deionized water at a concentration of 200 mg/mL for three hours. Then, CS (acetic acid 1%), PVA, and glycerol (water) were mixed with n-BG and CEO dispersions according to Table 1. Finally, the obtained mixture was taken to an ultrasonic bath (Branson, Madrid, Spain) to eliminate the bubbles in the solution. Then, the varieties were transferred onto glass molds in a preheated oven at 40 °C ± 0.2 to synthesize the CS/PVA/CEO/GLY/n-BG composites.

### 2.5. Characterization of CS/PVA/CEO/GLY/n-BG Composites

#### 2.5.1. X-ray Diffraction (XRD) and Fourier Transform Infrared Spectroscopy (FTIR)

X-ray diffraction studies were performed using a PANalytical X0Pert PRO diffractometer (Malvern Panalytical, Jarman Way, Royston, UK) with a Cu anode and a wavelength of Kα1 (0.154 nm). The diffraction 2θ angles were between 5–50°, with a scan speed of 2.63 s, a scan rate of 2 degrees/min, and a step size of 0.01°.

On the other hand, attenuated total reflectance–Fourier transform infrared spectroscopy (ATR-FTIR) was used to determine functional groups in the 500–4000 cm^−1^ range using an FT-IR-8400 instrument (Shimadzu, Kyoto, Japan) with a spectral resolution of 4 cm^−1^ and 32 scans.

#### 2.5.2. Thermal Analysis of the CS/PVA/CEO/GLY/n-BG Composites

For thermogravimetric analysis (TGA), we used a NETZSCH TG 209 F1 Libra instrument (Metter Toledo, Schwerzenbach, Switzerland). The samples were heated from 25–800 °C at 10 °C/min (nitrogen, 50 mL/min flow). The thermal transition of the pieces was studied under differential scanning calorimetry (DSC) (DSC1/500 instrument, Mettler Toledo, Schwerzenbach, Switzerland) using nitrogen, with 25–250 °C heating at 10 °C/min. The glass transition temperature (T_g_), melting temperature (T_m_), and crystallization temperature (T_c_) were calculated from the midpoint, endothermic, and exothermic peaks, respectively. The data were processed using TA Instruments Universal Analysis Software 2000 version 4.5A.

#### 2.5.3. Scanning Electron Microscopy

The morphology analysis of the CS/PVA/CEO/GLY/n-BG composite was run using a JEOL JSM-6490LA instrument (Musashino, Tokyo, Japan) in the secondary electron mode at 20 kV.

### 2.6. Surgical Preparation of Biomodels

The preliminary biocompatibility of the four composite formulations was assessed through the subcutaneous implantation method in a biomodel (five-month-old male Wistar rats, 380 g, Intermediate Laboratory of Preclinical Research and Biotherium, LABBIO, Universidad del Valle), as previously reported [22,35].

For the macroscopic examination and trichotomy, the samples were preserved in plastic containers after formalin and alcohol dehydration. Then, a xylol diaphanization and infiltration with paraffin using the Auto-technicon Tissue Processor™ from Leica Microsystem (Mannheim, Germany) process was utilized. The Thermo Scientific™ Histoplast Paraffin™ kit, provided by Fisher Scientific in Waltham, MA, USA, facilitated the cutting process of the samples for histology studies.

#### Histological Analysis

Paraffin blocks with the immobilized tissues were cut at a 5 µm thickness with a Leica microtome instrument. The histology studies used the hematoxylin–eosin (HE) and Masson’s trichotomy (MT) staining techniques. A Leica DFC 295 camera and Leica DM750 optical microscope were used for the image analysis with the Lecia Microsystems 4.12.0 software (Leica Microsystems, Mannheim, Germany).

The Biomedical Experimentation Animal Ethics Committee (CEAS) at the Universidad del Valle provided ethical approval for the study, which adhered to the guidelines established by the “Animal Research: Reporting of In Vivo Experiments” (ARRIVE) guidelines [36]. 

## 3. Results

### 3.1. Characterization of n-BG

Our previous work reported the synthesis and characterization of the n-BG 45S5 type [21].

### 3.2. CEO Characterization and Biological Activity

The GC-MS chemical composition of the CEO using the NIST and Wiley databases was based on the molecular weight and retention index. The chemical composition is summarized in Appendix A. After the CG-MS characterization of the CEO, 15 components were identified, with eugenol (76.5%), eugenyl acetate (17.8%), and *trans*-β-Caryophellene (4.0%) as the main components. Several authors have reported that essential oil compositions are usually influenced by age, soil composition, plant organs, edaphic conditions, genetic, extraction, and condition methods [25]. *S. aromaticum* L. contains approximately ~20% EO, classified by the Food and Drug Administration (FDA) as safe (GRAS) and used in several cosmetic and food goods applications. The main component of CEO has been identified as eugenol (at least 50%), but eugenyl acetate, β-caryophyllene, and α-humulene account for the remaining 10–40%, with different biological activities reported for each [26,37,38]. CEO has exhibited broad antimicrobial activity based on the high hydrophilic group content. Despite being antibacterial and food-pathogen-sensitive, fungi have also shown inhibition upon contact with CEO [39]. It is also essential to highlight that CEO presents antioxidant activity, which inhibits reactive essential oil (REO), preventing diseases such as cancer, arteriosclerosis, and Alzheimer’s, among others [39,40]. Also, CEO possesses anti-inflammatory and wound-healing activities, which are crucial for biomedical applications [27,40]. The anti-inflammatory effect of CEO is related to antioxidant activity that is mainly attributed to eugenol, a promising result that avoids synthetic drugs’ secondary effects [25]. There is also an analgesic-recognized effect of CEO, which is essential for regenerative medicine and wound healing [28].

### 3.3. FT-IR Spectroscopy

Figure 1 shows the characteristic bands of CS/PVA/CEO/GLY/n-BG composites to identify functional groups. In general, the formulations present typical bands attributed to the scaffold composition. It was possible to observe symmetric tension bands at 3321 cm^−1^ related to the -OH groups present in both PVA and CS [41]. Likewise, the symmetric and asymmetric tension band of the alkyl groups CH and CH_2_ at 2940 cm^−1^ was observed. In addition, the tension band of the carbonyl group (C=O) at 1659 cm^−1^ was related to the amido bonds I and II present in the CS [42,43], and finally, the symmetric tension band of the aliphatic C-O-C ester group at 1038 cm^−1^, as well as the flexural band of the C=O group at 1100 cm^−1^, were elucidated. Minor changes were observed concerning the formulations, especially in formulations F3 and F4, which contained CEO in their formulations. In this sense, variations in the intensity of the C=C band located at 1560 cm^−1^ are probably due to the presence of eugenol, the main component of CEO [44,45]. In addition, we observed that the addition of CEO and n-BG did not alter the bands related to CS and PVA bonds, which shows that their dispersion within the polymeric matrix was relatively homogeneous.

### 3.4. X-ray Diffraction

The different diffraction peaks of the CS/PVA/CEO/n-BG/GLY composites are shown in Figure 2. The diffractogram indicates that the amorphous nature of the materials predominates, with the appearance of small crystalline peaks. These peaks represent the PVA’s monocyclic unit cells, which are reflected at 2θ displacements of 19.7 and 22.2, attributed to the 110 and 200 planes, respectively. Also, new diffraction peaks with 2θ values of 32.1 and 40.3 were observed in the cases of F2 and F4, which could have originated due to the highly amorphous nature of the membranes. In addition, the considerable broadening and intensification of the peaks were related to the presence of CS, as observed in different works [46]. On the other hand, after the incorporation of CEO in formulations F3 and F4, the intensity of the diffraction peaks with 2θ shifts of 19.7 and 22.2 decreased, and the amplitude increased, indicating the alteration of the crystalline structure of the composite film. This observation is in agreement with works that have been carried out with the introduction of essential oils [47].

### 3.5. Thermal Analysis of CS/PVA/CEO/GLY/n-BG Composites

TGA allows the analyst to provide information about the formulation according to the material’s thermal properties. For this purpose, the thermal degradation by weight caused by increasing temperature was evaluated. Figure 3 shows all CS/PVA/CEO/GLY/n-BG composite thermograms and their respective derivatives. In these thermograms, no significant differences could be observed concerning the four degradation stages. The first degradation stage, between 60–100 °C, corresponds to the evaporation of partially absorbed weakly physically bound water. The second and third degradation stages, between 200–400 °C, are attributed to the disintegration of the PVA side chain and the related decomposition of the CS polysaccharide unit. Finally, the fourth stage of degradation occurred between 400–450 °C, corresponding to both the degradation of the glycosidic linkages of the CS and the degradation of the polyene residues from the PVA.

On the other hand, introducing glycerol and CEO to the F3 formulation (22.5%CS/70%PVA/5.0%CEO/2.5%GLY) was observed to decrease the thermal stability. In contrast to the F2 formulation (22.5%CS/70%PVA/5.0%n-BG/2.5%GLY), the addition of only Gly and n-BGs increased its stability compared to the control formulation, F1 (27.5%CS/70%PVA/2.5%GLY). This observation was because the introduction of n-BGs caused interactions between the hydroxyl groups found in the PVA, CS, and n-BGS components [48]; furthermore, thermal stability was attained because the silicate layers of the n-BGs acted as a barrier against degradative gases serving as thermal insulators, which was related to the hydrolysis conditions due to the polymeric components interfering with the heat transfer of these components [49,50]. However, the inclusion of CEO in the polymeric matrix composed of CS/PVA showed thermal strengthening as the weight loss decreased, i.e., while F1 showed a loss of 23.3%, F3 collected a residual mass of 18%. This phenomenon was probably due to a more robust film network being favored by the interaction between the polymeric matrix and the CEO, in addition to the plasticizing effect of the EO [51,52].

The different thermograms showing the melting temperature of chitosan (T_m1_) and PVA (T_m2_), as well as the crystallization temperature (T_cc_) attributed to n-BGs and the glass transition temperature (T_g_) for each of the formulations, can be observed in Figure 4. Table 2 summarizes the above thermal properties for each of the membranes.

DSC curves of the CS/PVA/CEO/GLY/n-BG composites show that CS was the predominant exothermic indicator of polymer degradation at around 280 °C. Concerning only PVA, the glass transition temperature was observed at approximately 48 °C without the presence of the peak attributed to the fusion of the semicrystalline structure of PVA at about 230 °C, which shows that the semicrystalline structure of PVA was not maintained in the different membranes. The small decompositions at lower temperatures were related to the elimination of water and the decomposition of the side chains of the polymers. On the other hand, it has been reported that the miscibility of the components was related to the intermediate glass transition temperature between the polymers [53]; in our case, it was around 50 °C, which confirms the excellent miscibility between the polymers involved. Additionally, no significant variations were found for T_g_, indicating that adding n-BG or CEO does not affect the amorphous part of the polymers and demonstrating the excellent affinity between the polymers.

On the other hand, formulation F2 presented a crystallization temperature of 341 °C, the exothermic peak present in the formulation. This observation was probably due to the incorporation of n-BGs, which offer thermally stable silicates in the presence of temperature [54].

### 3.6. Scanning Electron Microscopy (SEM) of CS/PVA/CEO/GLY/n-BG Composites

The analysis of the microstructure of the CS/PVA/CEO/GLY/n-BG composite can be observed in Figure 5. It is evident that formulations F1 and F3, which did not have n-BG, but did have the presence of glycerol, exhibited reasonably homogeneous, compact morphologies without breaks, thanks to the plasticizing effect of glycerol that facilitates the compact morphology of the chains [55]. Incorporating glycerol may have reduced the mobility of the CS and PVA chains, increasing their homogeneity and preventing cracking of the composites. However, F3 seemed to be more homogeneous due to the effect of the incorporation of CEO that could have reinforced the plasticizing effect of glycerol, inducing hydrogen bonds between CS and PVA through its oxygenated groups, given that eugenol is the main component of this oil (76.5%). This type of interaction between the components of ginger essential oil and polyesters has previously been observed [56].

On the other hand, incorporating n-BG into F2 and F4 generates greater roughness and some indentations and voids on the surfaces of the composites, which is favorable for cell adhesion and proliferation [57]. Furthermore, it has been shown that the biocompatibility of n-BG would facilitate interaction with cells, promoting the effects of pro-angiogenesis and osteogenesis [58]. 

### 3.7. Biocompatibility Analysis of CS/PVA/CEO/GLY/n-BG Composites

By the ISO 1093-6 standard (UNE-EN 30993-6) [59], once euthanasia was performed, a macroscopic inspection of the biomodels in the operated areas was carried out; Figure 6 corresponds to one of the biomodels that, after 60 days, showed hair recovery (Figure 6A). When trichotomy was performed, the skin appeared healthy, dry, and without evidence of lesions in the implantation area (Figure 6B). Likewise, after the incision and tissue separation, it was possible to see the four implantation zones on the inner surface of the skin (Figure 6C). There were no signs of necrosis, such as fistulas or purulent exudate, indicating a normal healing process.

#### 3.7.1. F1 Biocompatibility Study

When performing histological analyses, it was observed that, 60 days after implantation, the implanted material had a high rate of resorption and replacement by connective tissue, in addition to the presence of adipocytes. Using Gomori trichrome (GT) staining, the presence of type III collagen in the implantation area was evident, as was the absence of a fibrous capsule (Figure 7A,B). Figure 7B shows some remnants of the applied material and a significant accumulation of adipose tissue, indicating tissue recovery in the operated area. At a magnification of 100× and using Masson trichrome staining (MT), it was possible to identify adipocytes and remnants of the material in the middle of a network of type I collagen fibers (Figure 7C), indicating the recovery of the operated area with the absence of a fibrous capsule. The complete degradation and resorption of the implanted material and its replacement by connective tissue with the absence of a fibrous capsule can be explained by the biocompatibility of the three components of the formulation, as well as by the mechanical and biological properties of chitosan, which allows for rapid assimilation in the healing process [60,61,62].

In the subdermal implantation model (Figure 6), the area where the material is deposited corresponds to the hypodermis, a layer of tissue between the dermis and the muscle made up of connective tissue with a significant presence of adipocytes. Figure 7 shows the presence of connective tissue (Figure 7A) and many adipocytes (Figure 6C and Figure 7B), which indicates an almost complete recovery of the hypodermis. Fragments of the implanted material are not seen in the images.

#### 3.7.2. F2 Biocompatibility Study

In formulation F2, the decrease in the percentage of chitosan and the incorporation of n-BG affected the material resorption process. For this reason, after 60 days, large fragments of the composites in the process of degradation and resorption could be seen. Despite fragments of the material being observed, the healing process was carried out without a fibrous capsule in the implantation area. Like F1, this area appeared to be integrated with the rest of the tissue through type I collagen fibers from the muscle tissue (Figure 8A).

Figure 8A also shows that some areas where the material was reabsorbed were occupied by connective tissue (marked with a green star in the image), which can be better observed in Figure 8B. Here, the presence of type I collagen fibers separating a group of fragments from other groups and connective tissue in all spaces was also observed. In Figure 8C, at a magnification of 40×, particles in the process of resorption can be seen to be surrounded by inflammatory cells. Type III collagen fibers were also found surrounding the fragments. The incorporation of n-BG in this formulation could generate changes in healing as observed in the histological images, in which, despite large fragments of the composites persisting, there is evidence of advanced degradation and resorption of the material, as well as a process of replacement by tissue. Connective tissue, with integration to the neighboring tissue and the muscle, was observed with a network of type I and III collagen fibers inside the implantation area.

In the implantation area (Figure 8), fragments of the membranes of formulation F2 can be observed. In Figure 8A, type I collagen fibers, one of the skin’s main components, are seen in blue. In Figure 8B, at a magnification of 40×, it is possible to observe the presence of connective tissue (brown color) with fragments of the membranes in the process of degradation and numerous type I collagen fibers surrounding the fragments. In Figure 8C, these fragments can also be observed surrounded by type III collagen fibers (green), with multiple phagocytic cells attacking the nanocomposite fragments.

#### 3.7.3. F3 Biocompatibility Study

Formulation F3 differs from F2 in that the n-BG component was replaced by 5% CEO. An important histological finding was the presence of a fibrous capsule with parallel fibers surrounding the implantation area (Figure 9A). Using the Gomori trichrome technique, the presence of type III collagen fibers in the conformation of the capsule was identified (Figure 9A); Figure 9B,C show how the remnants of the implanted material were partially reabsorbed and replaced by a network of type I and type III collagen fibers, with the presence of adipocytes.

In Figure 9A, the black circle highlights the implantation area. There, what appears to be a connective tissue capsule composed of type III collagen fibers stands out. Membrane fragments are also immersed in a connective tissue matrix (brown). Figure 9B,C correspond to the area where fragments of the membranes persist. At a magnification of 40×, bundles of type I and type III collagen fibers from the capsule can be seen surrounding the membranes, with inflammatory cells phagocytosing the membranes.

#### 3.7.4. F4 Biocompatibility Study

As with the results of the previous implantation, after 60 days, a fibrous capsule delimiting the implantation area was observed, along with some fragments of the implanted material. Figure 10A shows that the capsule comprised type I collagen fibers, as identified by Masson’s trichrome staining, and there was an abundant presence of connective tissue inside the operated area, replacing the reabsorbed material.

Figure 10B corresponds to the interior of the implantation area; it is possible to see a fragment of remaining material surrounded by a capsule made up of type III collagen fibers. In Figure 10C, an image taken at a magnification of 100× makes it possible to observe the presence of blood vessels very close to the fragments of the material.

Figure 10A shows the implantation area surrounded by a capsule composed of collagen fibers, represented by blue in the image, and inside, a brown connective tissue matrix with some membrane fragments. Figure 10B indicates that each fragment, in turn, was surrounded by collagen fibers. In this case, the yellow circle indicates the presence of a capsule made up of type III collagen fibers. Figure 10C was taken at a magnification of 100×, showing the fragments surrounded by type III collagen fibers and numerous blood vessels.

The F4 formulations studied herein had CS, PVA, and Glycerol as standard components. CS is considered a biocompatible material with poor mechanical properties [63] but excellent biological properties, including the ability to accelerate the tissue-healing process [64], which would explain the rapid reabsorption of the material and its replacement by connective tissue. This could lead to an almost complete recovery of tissue architecture. On the other hand, PVA and glycerol are considered biocompatible and can improve the mechanical properties of CS.

In the studied formulations, the concentrations of PVA and glycerol always remained fixed; these are materials used as plasticizers that, in addition to improving the mechanical properties of the CS, are biocompatible [65]. However, changes in CS, bioglass, and CEO concentrations may be responsible for the different histological responses observed. Formulation F1 was the one that presented the fastest reabsorption and almost complete replacement by connective tissue at 60 days, as it was composed only of CS in a high percentage, PVA, and glycerol. The presence of connective tissue and aggregation of adipocytes indicates rapid tissue recovery by acquiring the regular appearance of tissue.

With the other formulations, it was observed that by decreasing the percentage of chitosan, the degradation and reabsorption process of the material became slower, which explains the presence of remnants of the material in the results observed for formulations F2, F3, and F4. The presence of remaining material after 60 days was also an indication of the stability of the composites. When using these composites as scaffolds, the degradation and resorption process must be carried out progressively. At the same time, wound healing occurs since it aims to stimulate tissue formation to recover tissue architecture, which is achieved through controlled degradation [66].

Histological results also change when new components are incorporated. In this way, integrating n-BG into F2 and F5 stimulated the formation of connective tissue with a different appearance than that observed in formulations F1 and F3. In general, n-BG have been widely used in scaffolds for tissue regeneration due to their bioactivity, which allows for the release of calcium and phosphorus [67]. This could cause different healing responses, as evidenced in Figure 9. The connective tissue seems to include tiny fragments of the material, while in Figure 8 and Figure 10, the connective tissue appears more homogeneous.

In Figure 9 and Figure 10, a fibrous capsule delimiting the implantation area was observed. This finding could be attributed to the presence of the CEO, because F3 and F4 were the only formulations in which CEO was included. When a material is subdermally implanted, a foreign body reaction is expected to form a fibrous capsule surrounding the material, which disappears when the material is degraded or reabsorbed by inflammatory cells [68]. In this work, no capsule formation was found due to the implantation of the first two formulations, unlike in formulations 3 and 4, which contained 5% CEO. These induced a slightly more intense inflammatory response when used in such high concentrations. 

Clove essential oils have been used as analgesics and antimicrobials in different applications [69,70]. It has also been reported that its anti-inflammatory effect can stimulate the re-epithelialization of wounds and rapid collagen formation with abundant collagen fibers [69]. Likewise, its potential as an antioxidant and antineoplastic agent is explored [71]. Depending on their concentrations, clove essential oils can stimulate an inflammatory response in the tissues [70,72], which could explain the presence of the fibrous capsule in the two formulations to which CEO was added.

Altogether, the histological results of the skin implantations show that the different formulations allowed the healing process to be carried out adequately. F1 is the formulation that presented the highest percentage of CS (27.5%), which induced a faster inflammatory response. It generated an almost complete degradation of the material in the absence of a fibrous capsule and with tissue recovery, as observed in Figure 7, where the restoration of the hypodermis with a deposit of adipocytes in a highly organized connective tissue matrix can be seen. 

The in vivo reabsorption of chitosan depends on several factors, such as molecular weight and the preparation method. Still, it is accepted that the inflammatory response is a mild and transient reaction to a foreign body. Some view chitosan as an inert biomaterial that induces no more than a gentle, brief foreign-body reaction [73]. 

By reducing chitosan and incorporating n-BG in F2, the inflammatory response changes, and a more significant presence of inflammatory infiltrate with less reabsorption of the material can be observed, but the fibrous capsule is still absent. Despite fragments of the material persisting, the absence of the capsule can be explained by the presence of the bioactive n-BG. This material is recognized for accelerating tissue healing by stimulating the more rapid formation of granulation tissue and a deposit. It also promotes accelerated collagen formation and angiogenesis [74].

In F3, the CS percentage of F2 was preserved, but the CEO replaced the n-BG. An interesting finding was the presence of a fibrous capsule that appeared to transform into scar connective tissue occupying the implantation area as the material was degraded and reabsorbed (Figure 9). In addition, adipocytes were observed, which may indicate the recovery of the hypodermis. It has been proposed that the CEO acts as an accelerator of healing, according to several results observed in animal models [27]. 

On the other hand, formulation F4 is the only one that contains CEO and n-BG. The implantation results show a faster resorption process than that of F3. Still, both present some features in common, such as the formation of highly organized connective tissue in the implantation area that seems to be formed at the expense of the fibrous capsule. It is thick and contains a large number of blood vessels. This early formation of an organized connective tissue can be explained by the effect of the release of ions from the n-BG with the capacity to positively affect the migration of epithelial cells, angiogenesis, and proliferation of fibroblasts [75].

In addition, as expected, CEO-containing membranes show the best signs of connective tissue regeneration and, therefore, a normal healing process. The above is probably because the addition of CEO through its main component (eugenol) promotes tissue healing through a foreign body mechanism. Several studies have shown that eugenol promotes protective action on liver cirrhosis by preventing liver cancer characterized by abnormal cell proliferation and decreasing oxidative stress, which could be the protective mechanism against liver cirrhosis [76]. On the other hand, it was demonstrated that a 2% gel based on CEO showed anti-inflammatory activity in the healing of decubitus wounds (pododermatitis) in rabbits [77].

## 4. Conclusions

In this study, we synthesized four membranes comprising CS/PVA/CEO/GLY/n-BG composites, which exhibited enhanced thermal stability compared to their components. CEO and n-BGs were confirmed through various characterization methods such as FTIR, XRD, TGA, and DSC. In FTIR analysis, we observed a sharp increase in the bands in the 1200–700 cm^−1^ region corresponding to C=C bonds of the aromatic ring, mainly due to the contribution of eugenol present in the essential oil. Furthermore, we observed that the inclusion of n-BGs promoted increases in the thermal properties and influenced the heat-resistant properties of CEO, which was particularly evident in formulations F3 and F4. The morphological examination conducted using scanning electron microscopy (SEM) revealed that the incorporation of CEO resulted in smooth surfaces, in contrast to the porous morphologies observed with n-BGs. Subdermal implantation of the CS/PVA/CEO/GLY/n-BG nanocomposites was assessed in a biomodel for 60 days, revealing high biocompatibility in this tissue. The excellent healing, hair recovery, and tissue remodeling without any normal immune response upon material degradation and resorption, along with the raising of type III collagen fibers, blood vessels, and inflammatory cells, demonstrated the material’s biocompatibility. We believe that these results are promising for a safe and eco-friendly approach to regenerative medicine.

## Figures and Tables

**Figure 1 polymers-15-04595-f001:**
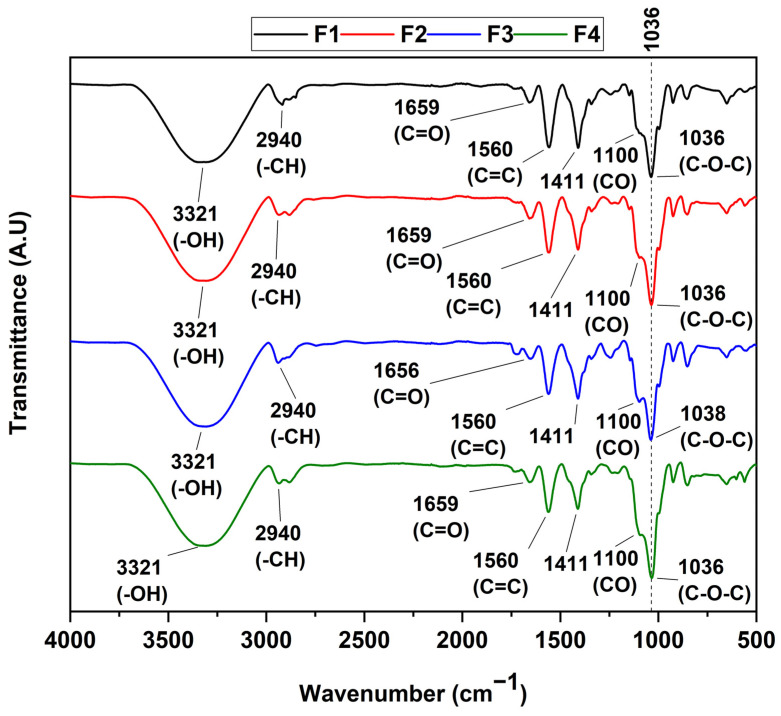
FT-IR spectrum of CS/PVA/CEO/GLY/n-BG composites.

**Figure 2 polymers-15-04595-f002:**
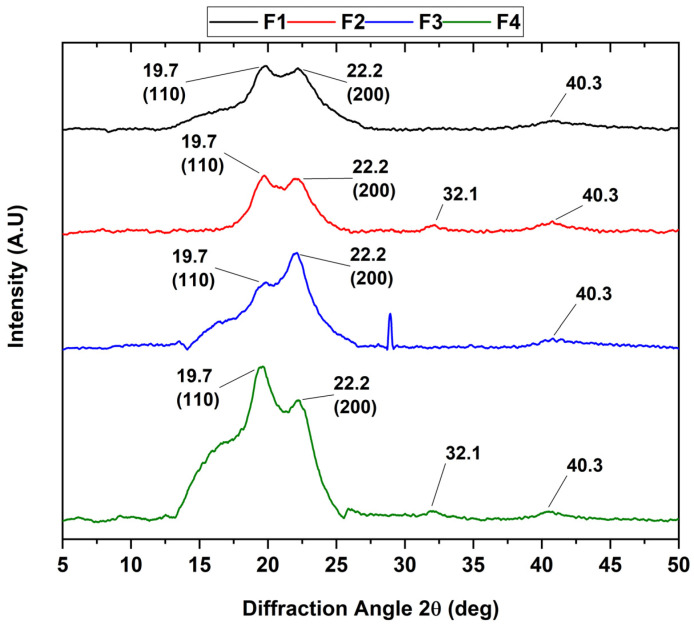
XRD analysis of CS/PVA/CEO/GLY/n-BG composites.

**Figure 3 polymers-15-04595-f003:**
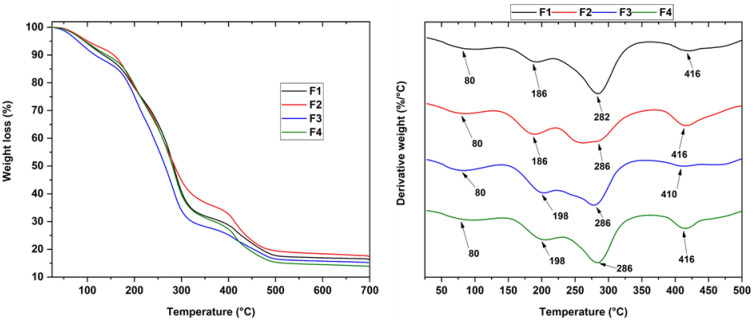
Thermogravimetric analysis of the CS/PVA/CEO/GLY/n-BG composites.

**Figure 4 polymers-15-04595-f004:**
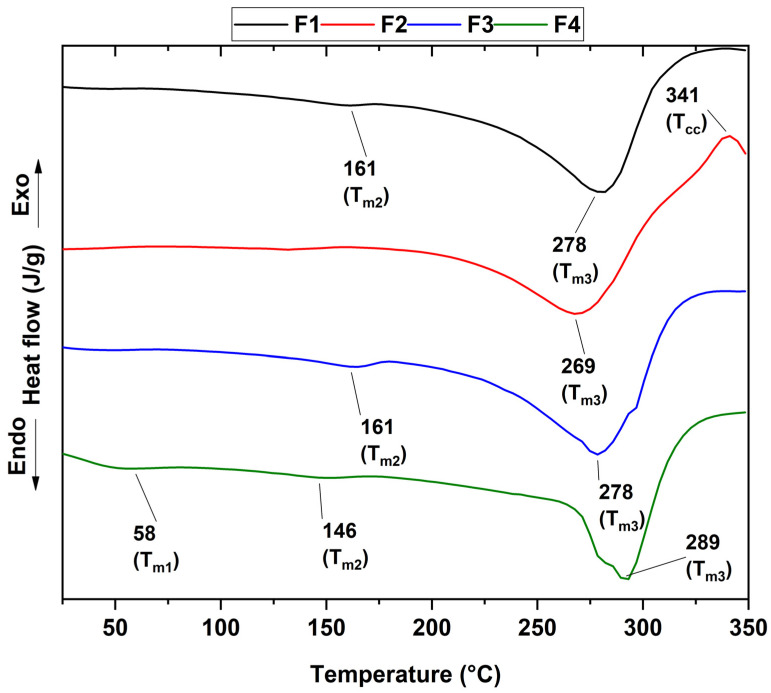
DSC curves of the CS/PVA/CEO/GLY/n-BG composites.

**Figure 5 polymers-15-04595-f005:**
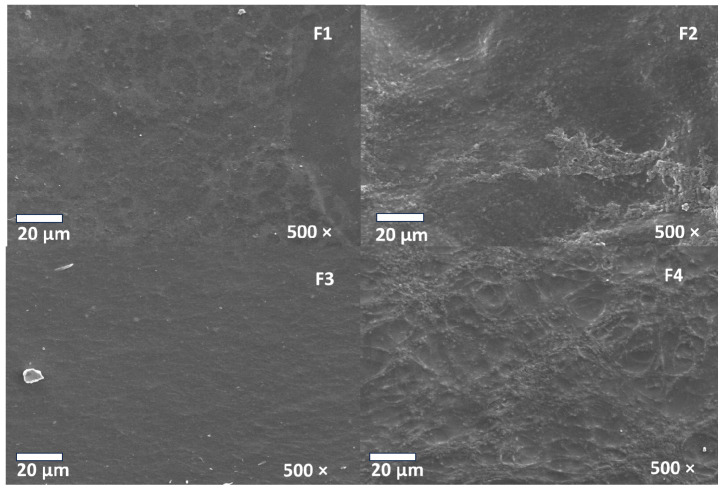
Morphology of membranes of CS/PVA/CEO/GLY/n-BG composites.

**Figure 6 polymers-15-04595-f006:**
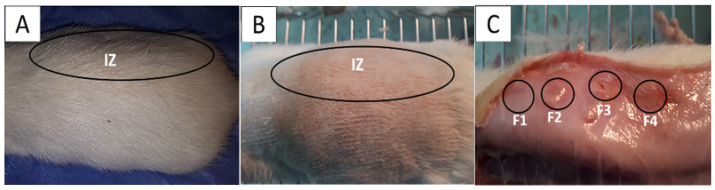
Subdermal implantation at 60 days on the dorsal surface of the Wistar rat. (**A**) Skin with hair recovery; (**B**): skin with trichotomy; (**C**): the inner surface of the skin. Ovals: Intervened area. IZ: Implantation area. Circles: areas of material implementation. Black circles: the implantation zones for the four formulations.

**Figure 7 polymers-15-04595-f007:**
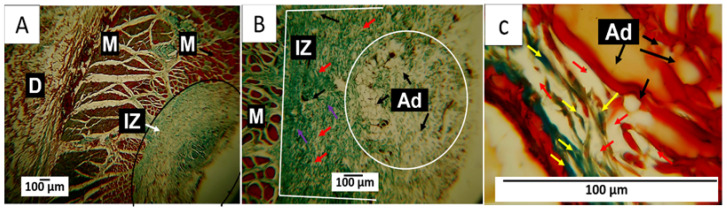
F1 formulation. Subdermal implantations at 60 days. (**A**) Image at 4×, GT technique. (**B**) Image at 10×, GT technique. (**C**) Image at 100×, MT technique. D: Dermis. M: Muscle. IZ: Implantation area. Black circle: Implantation area. Ad: adipocytes. White circle: Area with the presence of adipocytes. Red arrow: Fragments of the implanted material. Purple arrow: Type III collagen fibers. Yellow arrow: type I collagen fibers.

**Figure 8 polymers-15-04595-f008:**
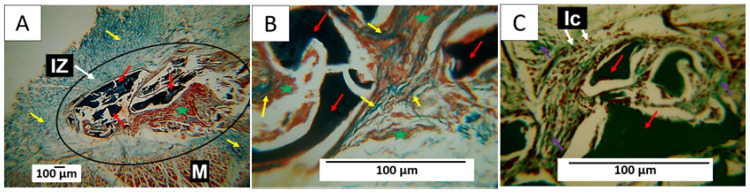
Formulation F2. Subdermal implantations at 60 days. (**A**) Image at 4×, MT technique. (**B**) Image at 40×, MT technique. (**C**) Image at 40×, GT technique. M: Muscle. IZ: Implantation area. Black circle: Implantation area. Yellow arrow: type I collagen fibers. Red arrow: Fragments of the implanted material. Purple arrow: Type III collagen fibers. Ic: Inflammatory cells. Green star: connective tissue.

**Figure 9 polymers-15-04595-f009:**
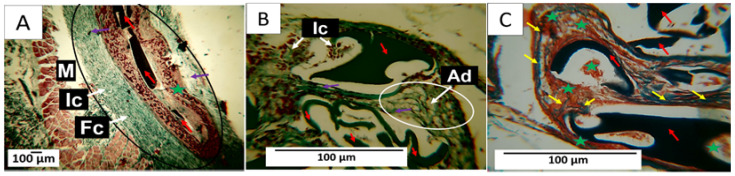
Formulation F3. Subdermal implantations at 60 days. (**A**) Image at 4×, GT technique. (**B**) Image at 40×, GT technique. (**C**) Image at 40×, MT technique. M: Muscle. Fc: Fibrous capsule. Black circle: Implantation area. Red arrow: Fragments of the implanted material. Purple arrow: Type III collagen fibers. Ic: Inflammatory cells. Yellow arrow: type I collagen fibers. Ad: adipocytes. Green star: connective tissue.

**Figure 10 polymers-15-04595-f010:**
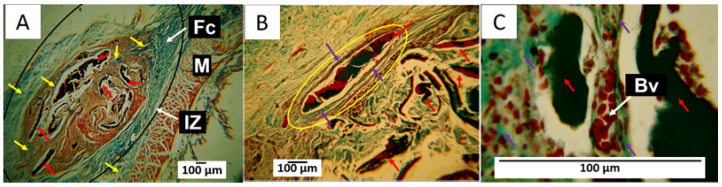
Formulation F4. Subdermal implantations at 60 days. (**A**) Image at 4×, MT technique. (**B**) Image at 10×, GT technique. (**C**) Image at 100×, GT technique. M: Muscle. IZ: Implantation area. Fc: Fibrous capsule. Black circle: Implantation area. Red arrow: Fragments of the implanted material. Yellow arrow: type I collagen fibers. Purple arrow: type III collagen fibers. Yellow circle: implantation area. Bv: blood vessels.

**Table 1 polymers-15-04595-t001:** Formulation of CS/PVA/CEO/GLY/n-BG composites.

Components	CS (%)	PVA (%)	CEO (%)	GLY (%)	n-BG (%)
F1	27.5	70	0	2.5	0
F2	22.5	70	0	2.5	5
F3	22.5	70	5	2.5	0
F4	17.5	70	5	2.5	5

**Table 2 polymers-15-04595-t002:** Thermal properties of the CS/PVA/CEO/GLY/n-BG composites.

	T_g_ (°C)	T_m1_ (°C)	T_m2_ (°C)	T_m3_ (°C)	T_cc_ (°C)
F1	48	-	161	278	-
F2	50	-	161	269	341
F3	51	-	-	278	-
F4	52	58	146	289	-

## Data Availability

Data will be made available upon request to the corresponding author.

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
