# Peer review of "Chitosan–Polyvinyl Alcohol Nanocomposites for Regenerative Therapy"

_polymers, 2023, doi:10.3390/polym15234595_

Round 1
Reviewer 1 Report
Comments and Suggestions for Authors
The Manuscript entitled “Chitosan-Polyvinyl Alcohol Nanocomposites for Regenerative Therapy” authored by Carlos David Grande-Tovar, Jorge Iván Castro, Diego López Tenorio, Paula A. Zapata, Edwin Florez-López, and Carlos Humberto Valencia-Llano presents an interesting work of research on the effect of clove essential oil in the purpose of wound healing. A few comments I would like to provide here to improve the information content of the manuscript before it can be accepted for publication and are listed below.
1. English language grammar and punctuation has to be corrected throughout the manuscript. E.g., Line 18, Line 28, Line 109, Line 182, line 207 (comma is missing), line 218, and so on.
2. Authors are requested to use the term carefully “Nano composite” or “nanostructured composite”.
The term nanostructured composite defines the case better.
3. In the FTIR analysis (Figure 1) the bending/ stretching vibrational modes are not clear form the figure.
4. In the XRD analysis (Figure 2) the indexing can be provided in terms of crystallographic planes and not in terms of 2Theta angles. Accordingly Figures 3 and 4 can be improved.
5. Figures 6-10 are impressive. The associated descriptions can be elaborated a little further.
6. The compositions of the formulations contain CEO which is a mixture of ingredients. In this regards authors can present a description on the relative merits of the individual compounds present in the CEO in the purpose of wound healing, If possible present a brief test case.
7. Overall I recommend a minor revision of the manuscript.
Comments on the Quality of English Language1. English language grammar and punctuation has to be corrected throughout the manuscript. E.g., Line 18, Line 28, Line 109, Line 182, line 207 (comma is missing), line 218, and so on.
Author Response
- English grammar and punctuation have to be corrected throughout the manuscript. E.g., Line 18, Line 28, Line 109, Line 182, line 207 (comma is missing), line 218, and so on.
R// We appreciate the reviewer’s comments. The manuscript was thoroughly reviewed and corrected for English grammar.
- Authors are requested to use the term carefully “Nano composite” or “nanostructured composite”.
R// We appreciate it very much. We corrected the whole text with the word nanocomposite since the definition offered by Chen et al. (2007), “Nanocomposites can be defined as multicomponent materials comprising multiple different (nongaseous) phase domains in which at least one type of phase domain is a continuous phase and in which at least one of the phases has at least one dimension of the order of nanometers” [1]. (Line 94)
- Anand, M.; Sathyapriya, P.; Maruthupandy, M.; Beevi, A.H. Synthesis of Chitosan Nanoparticles by TPP and Their Potential Mosquito Larvicidal Application. Front. Lab. Med. 2018, 2, 72–78.
- In the FTIR analysis (Figure 1), the bending/ stretching vibrational modes are not clear from the figure.
R// We appreciate the suggestion. The following correction in the FT-IR section was introduced (lines 214-228).
Figure 1 shows the characteristic bands of CS/PVA/CEO/GLY/n-BG composites to identify functional groups. In general, the formulations present typical bands attributed to the scaffold composition. In this order of ideas, it was possible to observe symmetric tension bands at 3321 cm-1 related to the -OH groups present in both PVA and CS [2]. Likewise, the symmetric and asymmetric tension band of the alkyl groups CH and CH2 at 2940 cm-1 were observed; in addition, the tension band of the carbonyl group C=O at 1659 cm-1 related to the amido bonds I and II present in the CS was observed [3,4]. Finally, the symmetric tension band of the aliphatic C-O-C ester group at 1038 cm-1 and the flexural band of the C=O group at 1100 cm-1 were elucidated. Minor changes were observed concerning the formulations, especially in formulations F3 and F4, which contain CEO in their formulation. In this sense, variations in the intensity of the C=C band located at 1560 cm-1 are probably due to the presence of eugenol, the main component of CEO [5,6]. In addition, it is observed that the addition of CEO and n-BG did not alter the bands related to CS and PVA bonds, which shows that their dispersion within the polymeric matrix was relatively homogeneous.
- In the XRD analysis (Figure 2) the indexing can be provided in terms of crystallographic planes and not in terms of 2Theta angles. Accordingly, Figures 3 and 4 can be improved.
R// Thank you very much for the suggestion. Due to the amorphous nature of the membranes intrinsically related to the CS component, it is impossible to observe crystallographic peaks for this component. However, due to the PVA's semi-crystalline nature, they diffract and can be seen in the figure at a specific 2θ distance with their corresponding diffraction plane. Concerning the new diffraction peaks, these are related to the natural amorphous nature of the membrane and, therefore, do not have diffraction planes related to the components that constitute the membranes. Figures 3 and 4 were also improved, especially Figure 4, where we included the type of transition in each temperature.
- Figures 6-10 are impressive. The associated descriptions can be elaborated a little further.
R// We appreciate the reviewer’s comment. We added more explanation between lines 360-364, 390-395, 410-415, and 432-437.
- The compositions of the formulations contain CEO which is a mixture of ingredients. In this regards authors can present a description on the relative merits of the individual compounds present in the CEO in the purpose of wound healing, If possible present a brief test case.
R// We appreciate the reviewer’s comment. We added more explanation between lines 486 and 517.
“Altogether, the histological results of the skin implantations show that the different formulations allowed the healing process to be carried out adequately. F1 is the formulation that presents the highest percentage of CS (27.5%), which induced a faster inflammatory response, generating an almost complete degradation of the material in the absence of a fibrous capsule and with tissue recovery, as observed in Figure 7, where the restoration of the hypodermis with a deposit of adipocytes in a highly organized connective tissue matrix can be seen.
The in vivo reabsorption of chitosan depends on several factors, such as molecular weight and the preparation method. Still, it is accepted that its inflammatory response is a mild and transient reaction to a foreign body. Some view chitosan as an inert biomaterial that induces no more than a gentle, brief foreign-body reaction [72].
By reducing chitosan and incorporating n-BG in F2, the inflammatory response changes, observing a more significant presence of inflammatory infiltrate with less reabsorption of the material, but the fibrous capsule is still absent. Despite fragments of the material persisting, the absence of the capsule is explained by the presence of the bioactive n-BG. This material is recognized for accelerating tissue healing by stimulating the more rapid formation of granulation tissue and a deposit. It also promotes accelerated collagen formation and angiogenesis [73].
In F3, the CS percentage of F2 is preserved, but the CEO replaces the n-BG. An interesting finding is the presence of a fibrous capsule that appears to have transformed into a scar connective tissue that occupies the implantation area as the material is degraded and reabsorbed (Figure 9). In addition, adipocytes are observed, which may indicate the recovery of the hypodermis. It has been proposed that the CEO acts as an accelerator of healing, according to several results observed in animal models [27].
On the other hand, formulation F4 is the only one that contains CEO and n-BG. The implantation results show a faster resorption process than that of F3. Still, both present some features in common, such as the formation of a highly organized connective tissue in the implantation area that seems to be formed at the expense of the fibrous capsule. Thick and contains a large number of blood vessels. This early formation of an organized connective tissue can be explained by the effect of the release of ions from the n-BG with the capacity to act positively on the migration of epithelial cells, angiogenesis, and proliferation of fibroblasts [74].
In addition, as expected, CEO-containing membranes show the best signs of connective tissue regeneration and, therefore, a normal healing process. The above is probably because the addition of CEO through its main component (Eugenol) promotes tissue healing through a foreign body mechanism. Several studies have shown that Eugenol promotes a protective action on liver cirrhosis by preventing liver cancer characterized by abnormal cell proliferation and decreasing oxidative stress, which could be the protective mechanism against liver cirrhosis [75]. On the other hand, it was demonstrated that a 2% gel based on CEO obtained an anti-inflammatory activity in the healing of decubitus wounds (pododermatitis) in rabbits [76].

Reviewer 2 Report
Comments and Suggestions for Authors
This work presents a study of the effect of Clove Oils and nanobioglass on the behavior of biocomposites membranes. The membranes were subjected to experimental tests to analyze their thermal stability, FT-IF spectrum, morphology, and biocompatibility. The examination of the implanted membranes demonstrated their biocompatibility and biodegradability. The paper is well written an organized. Only the introduction must be improved before considering this paper for publication.
The introduction is mainly focused on the description of the biomaterials used in each published paper. But the results of each paper are not analyzed. Author should explain the influence of different components that other authors found on the thermal stability, biocompatibility and biodegradability of the different biomaterials analyzed. These results can help to get a better understanding of the present results.
Author Response
The introduction is mainly focused on the description of the biomaterials used in each published paper. But the results of each paper are not analyzed. Author should explain the influence of different components that other authors found on the thermal stability, biocompatibility and biodegradability of the different biomaterials analyzed. These results can help to get a better understanding of the present results.
R// Thank you very much for the suggestion. In the introduction, we added two paragraphs indicating possible applications with the results obtained. Lines 84-90 and lines 109-113
Among the n-BGs that can be used for biomedical applications are 1393BG and 45S5, with a composition that binds different minerals such as SiO2, Na2O, CaO, P2O5, and others. However, the most used is 45S5 because its composition does not allow the production of necrotic tissue and pus
production [7,8]. The main disadvantage is the adequate control of porosity, which requires specific conditions; therefore, these conditions can affect the nucleation phenomena ideal for biomedical procedures [9–11].
Pd nanoparticles have been made together with the CEO to be evaluated as a cytotoxic agent against cervical cancer, where it was demonstrated that this material is biocompatible and induced a concentration-dependent inhibition against HeLA cells that could be due to the pharmacologically active compounds retained on the surface of the PdNPs of the clove bud extract [12].
